# Enhancing Pruned Models by Input Compensation

## Abstract

Although foundation models are powerful, they are large and require substantial memory and computation resources to serve. To address this issue, many pruning methods have been proposed to reduce model size, thereby achieving memory and computational efficiency. These methods *adjust the retained weights* to compensate for the removed weights. In this paper, we propose a novel approach called input compensation (IC) to improve the performance of pruned models, i.e., *adjust the input* to compensate for the removed weights. Unlike existing pruning methods, which are designed in the parameter space, the proposed IC is designed in the input space. Hence, IC is complementary to existing methods and can be integrated with them. Extensive experiments on various tasks, including image classification, language modeling, and image generation, demonstrate that IC effectively boosts the performance of pruned models.

## 1. Introduction

Foundation models (Radford et al., 2021; Touvron et al., 2023b; Podell et al., 2024) have achieved great success in a variety of domains, such as computer vision and natural language processing. As the availability of data and computational resources expands, these models have scaled in both size and performance (Touvron et al., 2023a;b; Meta, 2024). However, the substantial number of parameters in these models require extensive computational resources to serve, making it challenging to deploy them on resource-constrained devices such as smartphones and laptops. To reduce the costs, numerous *model compression* techniques have been proposed to reduce the model size, e.g., distillation (Polino et al., 2018; Wang et al., 2019; Liang et al., 2023), quantization (Lin et al., 2024; Dettmers et al., 2022; Shao et al., 2024; Xiao et al., 2023), and pruning (Han et al., 2015; Frantar & Alistarh, 2023; Zhang et al., 2024; Sun et al., 2024). Quantization requires specialized hardware support, while distillation requires extensive retraining. Hence, we focus on pruning, which is a simple and representative technique.

*Pruning* reduces the model size by removing individual weights or rows/columns according to their importance scores. A pruned model can achieve promising performance with fewer parameters, resulting in a noticeable reduction in memory and computational demands. A simple but effective pruning method is Magnitude Pruning (Han et al., 2015) which removes weights according to their magnitudes. The underlying assumption is that weights with smaller values contribute less to overall performance. However, this assumption does not always hold and many advanced methods (Sun et al., 2024; Frantar & Alistarh, 2023; Zhang et al., 2024) have been proposed recently.

Current state-of-the-art pruning methods (Frantar & Alistarh, 2023; Das et al., 2023; Zhang et al., 2024; Sun et al., 2024; Dong et al., 2024; An et al., 2024) focus on the *parameter space* to improve pruning efficacy and can be roughly categorized into two groups: (i) designing an effective score to measure the importance of weight, and (ii) adjusting the remaining unpruned weights to reduce the error caused by the pruned weights. For example, Wanda (Sun et al., 2024) designs an importance score to incorporate input activations with weight magnitude to take outlier features into consideration instead of only weight magnitudes in Magnitude Pruning; SparseGPT (Frantar & Alistarh, 2023) proposes to adjust the unpruned weights by minimizing reconstruction loss using the Optimal Brain Surgeon framework (Hassibi et al., 1993; Singh & Alistarh, 2020; Frantar et al., 2021). The pruned model can be formulated as $\mathcal{F}(\mathbf{X}; \mathbf{W} \odot \mathbf{M} + \boldsymbol{\Delta}_{\mathbf{w}})$, where $\mathcal{F}$ is the model, $\mathbf{X}$ is the input, $\mathbf{W}$ is the weight matrix, $\mathbf{M}$ is the weight mask determined by the importance score, $\odot$ is element-wise multiplication, and $\boldsymbol{\Delta}_{\mathbf{w}}$ (called *weight compensation*) is an update matrix for the retained weights.

In this paper, we propose a novel method, called input compensation (IC), for enhancing pruned models by adjusting the input to compensate for the removed weights. Specifically, we learn an input compensation $\boldsymbol{\Delta}_{\mathbf{x}}$ to adjust the input $\mathbf{X}$, therefore the output of the pruned model is determined by $\mathcal{F}(\mathbf{X} + \boldsymbol{\Delta}_{\mathbf{x}}; \hat{\mathbf{W}})$, where $\hat{\mathbf{W}}$ is a sparse weight matrix corresponding to the pruned model. We learn a compensation pool consists of multiple candidate compensations and $\boldsymbol{\Delta}_{\mathbf{x}}$ is a weighted combination of the candidate compensations via the attention mechanism (Vaswani et al., 2017).

Different from existing pruning methods, the proposed IC is designed in the *input space*. Hence, IC complements existing methods that operate in the parameter space and can be integrated with them to boost their performance. Extensive experiments on computer vision and natural language processing show that IC brings a large improvement to existing pruning methods.

Our contributions are summarized as follows: (i) We propose IC which is a novel direction to enhance pruned models; (ii) IC is designed in the input space and, thus, is orthogonal to existing pruning methods designed in the parameter space. Hence, IC can be combined with existing pruning methods; (iii) Experimental results on various tasks demonstrate that IC is beneficial to existing pruning methods.

## 2. Related Work

**Model Compression.** Foundation Models (Touvron et al., 2023a;b; Meta, 2024; Ho et al., 2020; Radford et al., 2021) are large pre-trained models designed to serve as base models for various downstream tasks. Though foundation models are powerful, their massive parameters usually require extensive computational and memory resources. Many recent efforts have been devoted to reducing the cost via model compression (Frantar & Alistarh, 2022; Xu et al., 2024; Wang et al., 2024). The most popular methods for model compression are pruning, quantization, and distillation. *Pruning* (Han et al., 2015; Zhang et al., 2024; Sun et al., 2024; Dong et al., 2024; Das et al., 2023; An et al., 2024; Frantar & Alistarh, 2023) discards parts of the model that are less important or redundant. *Quantization* (Lin et al., 2024; Dettmers et al., 2022; Shao et al., 2024; Xiao et al., 2023; Yao et al., 2022; Kim et al., 2024) is a technique to reduce the computational complexity and memory footprint of a neural network by converting the model's parameters (weights and activations) from higher-precision representations (such as 32-bit floating-point) to lower-precision ones (such as 8-bit integers). The primary goal of quantization and pruning is to make the model more compressed without significantly sacrificing its performance. *Distillation* (Polino et al., 2018; Wang et al., 2019; Liang et al., 2023) trains a smaller and more efficient model to replicate the behavior of a larger and more complex model, thereby retaining much of its performance while significantly reducing computational resources. Quantization demands specialized hardware (e.g., NVIDIA TensorRT) that supports lower precision arithmetic, while distillation requires an expensive training phase to transfer knowledge from a large teacher model to a small student model. In this paper, we focus on pruning, which is a simple and widely used approach.

**Pruning** aims to remove less important weights without significant performance degradation. Several important met-

rics have been designed recently. The simplest one is based on the parameter magnitude, i.e., Magnitude Pruning (Han et al., 2015). Wanda (Sun et al., 2024) further incorporates weight magnitude with their input activations to consider outlier features when calculating importance scores, while RIA (Zhang et al., 2024) uses relative importance as a pruning metric. Taylor pruning (Molchanov et al., 2022) designs a score based on the weight multiplied by its gradient, while Diff-Pruning (Fang et al., 2023) further uses Taylor expansion over pruned timesteps to identify and discard unimportant parameters. In addition to designing importance scores to find less useful parameters, one can *update the unpruned weights* to compensate for the error caused by the pruned weights. For example, SparseGPT (Frantar & Alistarh, 2023) and OBC (Frantar & Alistarh, 2022) propose to update the unpruned weights by minimizing a reconstruction loss by the Optimal Brain Surgeon framework (Hassibi et al., 1993; Singh & Alistarh, 2020; Frantar et al., 2021). Different from SparseGPT and OBC, we propose input compensation by *adjusting the inputs* to reduce the error caused by pruning.

**Prompting** (Radford et al., 2019; Brown et al., 2020; Liu et al., 2022; Ding et al., 2022) is a popular method used in transformer-based models which inserts additional tokens that instruct the model to generate a specific kind of response. These tokens can be either discrete tokens (e.g., "The topic is" for topic classification (Zhang et al., 2022a; Hou et al., 2022; Jiang et al., 2023), "Let's think step by step" for reasoning tasks (Kojima et al., 2022)) or learnable continuous vectors (e.g., prompt tuning (Lester et al., 2021; Liu et al., 2021; Zhang et al., 2022b) or prefix learning (Li & Liang, 2021; Liu et al., 2023)). Unlike prompting that inserts extra tokens into the inputs, our input compensation edits the inputs directly. Furthermore, compensations are input-dependent, while prompts are usually input-independent (Ding et al., 2022; Lester et al., 2021; Liu et al., 2021; Zhang et al., 2022b; Bahng et al., 2022).

In control systems, the idea of **input compensation** (Kuo & Golnaraghi, 1995; Franklin et al., 2002) is practically used to adjust the control signal to reduce the influence of disturbance. The goal is to adjust the input so that the overall system achieves the desired behavior, such as better stability, faster response, or improved accuracy. For example, in feedforward compensation (Campos & Lewis, 1999; Krstic, 2009), if a disturbance is known ahead of time (e.g., wind gusts affecting an airplane), this information can be incorporated into the control signal so that the system compensates for it before it affects the output. In model pruning, the pruned weights can be viewed as disturbances and we use input compensation to enhance pruned models.

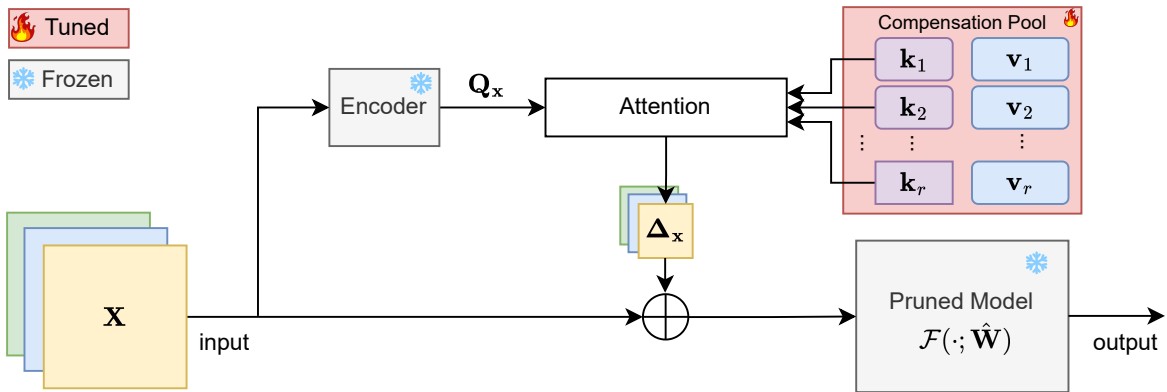

Figure 1: Input compensation for pruned models.

## 3. Preliminary on Pruning Models

Let $\mathbf{W} \in \mathbb{R}^{d_i \times d_o}$ be a weight matrix of a model $\mathcal{F}$ and $\mathbf{S}$ be a scoring matrix whose $\mathbf{S}_{i,j}$ measures the importance of $\mathbf{W}_{i,j}$. To prune $p\%$ parameters of $\mathbf{W}$, a threshold $\beta$ is determined to satisfy $\frac{\#\{\mathbf{S}_{i,j}:|\mathbf{S}_{i,j}|<\beta\}}{\#\{\mathbf{S}_{i,j}\}} = p\%$. Using the threshold, we construct a binary weight mask $\mathbf{M}$ whose $\mathbf{M}_{i,j} = 1$ if $|\mathbf{S}_{i,j}| \geq \beta$ else 0 and prune the model as $\mathbf{W} \odot \mathbf{M}$. To improve the performance of the pruned model, one can adjust the unpruned weights to compensate for the removed weights. Generally, the pruned model is formulated as:

$$\mathcal{F}(\mathbf{X}; \mathbf{W} \odot \mathbf{M} + \boldsymbol{\Delta}_{\mathbf{w}}), \qquad (1)$$

where $\boldsymbol{\Delta}_{\mathbf{w}}$ (called *weight compensation*) is an update matrix for the retained weights. Various methods have been proposed to enhance pruning by designing an effective scoring metric or learning an effective weight compensation $\boldsymbol{\Delta}_{\mathbf{w}}$, e.g., Han et al. (2015); Zhang et al. (2024); Sun et al. (2024); Dong et al. (2024); Das et al. (2023); An et al. (2024) for the former, and Frantar & Alistarh (2023; 2022) for the latter. We briefly review three representative pruning approaches.

**Magnitude Pruning** (Han et al., 2015) is the simplest technique whose score matrix is defined as $\mathbf{S}_{i,j} = |\mathbf{W}_{i,j}|$, i.e., removing the weights whose magnitudes are below a predefined threshold. In practice, magnitude pruning is performed layer-wise: for each layer, a layer-dependent threshold is determined based on the local distribution of weights. Though Magnitude pruning has stood out as a strong baseline for pruning models (Blalock et al., 2020), it has a major limitation: it ignores the importance of input activation, which plays an equally importance role as weight magnitudes in determining the output of linear layers (e.g., fully connected layers, attention layers).

**Wanda** (Sun et al., 2024) addresses this limitation by incorporating both weights and inputs into defining the weight importance. Specifically, let $\mathbf{X} \in \mathbb{R}^{N \times d_i}$ (where $N$ is the

sequence length) be the input activation of a training sample. Consider a linear layer $\mathbf{Y} = \mathbf{XW}$, Wanda defines the importance of $\mathbf{W}_{i,j}$ as $\mathbf{S}_{i,j} = |\mathbf{W}_{i,j}| \cdot \|\mathbf{X}_{:,i}\|_2$.

**SparseGPT** (Frantar & Alistarh, 2023) introduces a more sophisticated pruning approach by incrementally pruning each column of $\mathbf{W}$, followed by adjusting the remaining weights to compensate for those that have been pruned by the Optimal Brain Surgeon framework (Hassibi et al., 1993; Singh & Alistarh, 2020; Frantar et al., 2021). The score matrix is determined by $\mathbf{S}_{i,j} = \frac{|\mathbf{W}_{i,j}|^2}{[\mathbf{H}^{-1}]_{i,i}}$ and $\mathbf{H} = \mathbf{X}^{\top}\mathbf{X} + \lambda \mathbf{I}$ ($\lambda$ is a small positive constant) is the Hessian matrix of the reconstruction loss.

## 4. Methodology

Different from existing methods, which primarily focus on enhancing pruning in the *parameter* space, e.g., learning a good scoring metric $\mathbf{S}$ or weight compensation $\boldsymbol{\Delta}_{\mathbf{w}}$, we propose enhancing pruning in the *input* space. Let $\hat{\mathbf{W}}$ be the weight of a pruned model. Our objective is to *determine an input compensation $\boldsymbol{\Delta}_{\mathbf{x}}$ for the input such that the output of the pruned model approximates that of the dense model*, i.e.,

$$\mathcal{F}(\mathbf{X} + \boldsymbol{\Delta}_{\mathbf{x}}; \hat{\mathbf{W}}) \approx \mathcal{F}(\mathbf{X}; \mathbf{W}). \qquad (2)$$

The compensation $\boldsymbol{\Delta}_{\mathbf{x}}$ depends on the input $\mathbf{X}$. Obviously, learning $\boldsymbol{\Delta}_{\mathbf{x}}$ from scratch for each input is inefficient. To deal with this issue, we begin by developing a framework of learning input compensation within the context of a simple linear layer and subsequently extend this approach to general models.

### 4.1. Linear Models

Recent studies (Yu et al., 2017; Li et al., 2023b; Ding et al., 2023) have shown that the weight matrix $\mathbf{W}$ of neural networks can be approximated by a combination of a sparse

matrix $\mathbf{S} \in \mathbb{R}^{d_i \times d_o}$ (assume $\text{rank}(\mathbf{S}) = d_o$) and a low-rank matrix $\mathbf{A}\mathbf{B}^\top$ (where $\mathbf{A} \in \mathbb{R}^{d_i \times r}$ and $\mathbf{B} \in \mathbb{R}^{d_o \times r}$, $r$ is the rank). Hence, for a linear layer, its output can be approximated as:

$$\mathbf{Y} = \mathbf{X}\mathbf{W} \approx \mathbf{X}(\mathbf{S} + \mathbf{A}\mathbf{B}^\top) = \mathbf{X}\mathbf{S} + \mathbf{X}\mathbf{A}\underbrace{\mathbf{B}^\top(\mathbf{S}^\top\mathbf{S})^{-1}\mathbf{S}^\top}_{\equiv \hat{\mathbf{B}}}\mathbf{S}$$

$$= \left(\mathbf{X} + \underbrace{\mathbf{X}\mathbf{A}\hat{\mathbf{B}}^\top}_{\text{i.e., }\boldsymbol{\Delta}_\mathbf{x}}\right)\mathbf{S}. \quad (3)$$

Note that input compensation and weight compensation are dual for linear models, i.e., for an input $\mathbf{X}$ and its compensation $\boldsymbol{\Delta}_\mathbf{x}$, one can design a weight compensation $\boldsymbol{\Delta}_\mathbf{w} = \mathbf{X}^\top(\mathbf{X}\mathbf{X}^\top)^{-1}\boldsymbol{\Delta}_\mathbf{x}\mathbf{W}$ (thus, $\mathbf{X}\boldsymbol{\Delta}_\mathbf{w} = \boldsymbol{\Delta}_\mathbf{x}\mathbf{W}$) such that

$$(\mathbf{X} + \boldsymbol{\Delta}_\mathbf{x})\mathbf{W} = \mathbf{X}\mathbf{W} + \boldsymbol{\Delta}_\mathbf{x}\mathbf{W} = \mathbf{X}(\mathbf{W} + \boldsymbol{\Delta}_\mathbf{w}). \quad (4)$$

However, for nonlinear models $\mathcal{F}(\mathbf{X}; \mathbf{W})$, the dual property does not hold, i.e., for an input compensation $\boldsymbol{\Delta}_\mathbf{x}$, there does not exist $\boldsymbol{\Delta}_\mathbf{w}$ such that $\mathcal{F}(\mathbf{X} + \boldsymbol{\Delta}_\mathbf{x}; \mathbf{W}) = \mathcal{F}(\mathbf{X}; \mathbf{W} + \boldsymbol{\Delta}_\mathbf{w})$.

Let $\mathbf{a}_i$ and $\hat{\mathbf{b}}_i$ be the $i$-th column of $\mathbf{A}$ and $\hat{\mathbf{B}}$, respectively. Then, according to (3), the $i$-th row of $\boldsymbol{\Delta}_\mathbf{x}$ can be re-expressed as $\sum_{j=1}^r (\mathbf{x}_i^\top \mathbf{a}_j)\hat{\mathbf{b}}_j$, which is similar to the attention mechanism (Vaswani et al., 2017): $\{\mathbf{x}_i\}$ are the query, $\{\mathbf{a}_j\}$ are the keys, and $\{\hat{\mathbf{b}}_j\}$ are the values. This observation inspires us to design a framework of IC for general models.

### 4.2. General Models

Building on the above insight from the linear layer, we propose a general framework based on the attention mechanism for general models. The proposed procedure is shown in Algorithm 1. Figure 1 provides an overview of the IC framework, consisting of a frozen encoder $\mathcal{E}(\cdot)$ and a learnable compensation pool $(\mathbf{K}, \mathbf{V})$ (where $\mathbf{K} \in \mathbb{R}^{d_e \times r}$ and $\mathbf{V} \in \mathbb{R}^{r \times d_i}$). The encoder, which can either be a small submodule of the pruned model or an identity function, maps $\mathbf{X}$ into an embedding $\mathbf{Q}_\mathbf{x} = \mathcal{E}(\mathbf{X}) \in \mathbb{R}^{N \times d_e}$, while the compensation pool consists of $r$ learnable compensations. Based on attention mechanism, the input compensation is then constructed as:

$$\boldsymbol{\Delta}_\mathbf{x} = \text{softmax}\left(\frac{\mathbf{Q}_\mathbf{x}\mathbf{K}}{\sqrt{d_e}}\right)\mathbf{V}. \quad (5)$$

We add $\boldsymbol{\Delta}_\mathbf{x}$ to the input and learn the compensation pool by minimizing the following supervised loss:

$$\min_{\mathbf{K},\mathbf{V}} \sum_{(\mathbf{X},\mathbf{Y})\in\mathcal{D}} \ell(\mathcal{F}(\mathbf{X} + \boldsymbol{\Delta}_\mathbf{x}; \hat{\mathbf{W}}), \mathbf{Y}), \quad (6)$$

---

**Algorithm 1** Input Compensation.

---

**Require:** pruned model $\mathcal{F}(\cdot; \hat{\mathbf{W}})$, training set $\mathcal{D}$, rank $r$, step size $\eta$, encoder $\mathcal{E}(\cdot)$;
1: initialize $\mathbf{K} \in \mathbb{R}^{d_e \times r}$ and $\mathbf{V} \in \mathbb{R}^{r \times d_i}$;
2: **for** $t = 1, \dots, T$ **do**
3:      sample an input $\mathbf{X}$ from $\mathcal{D}$;
4:      compute query embedding $\mathbf{Q}_\mathbf{x} = \mathcal{E}(\mathbf{X})$;
5:      compute compensation $\boldsymbol{\Delta}_\mathbf{x} = \text{softmax}\left(\frac{\mathbf{Q}_\mathbf{x}\mathbf{K}}{\sqrt{d_e}}\right)\mathbf{V}$;
6:      compute gradient $\mathbf{g} = \nabla_{(\mathbf{K},\mathbf{V})}\ell(\mathcal{F}(\mathbf{X}+\boldsymbol{\Delta}_\mathbf{x}; \hat{\mathbf{W}}), \mathbf{Y})$
     (or $\nabla_{(\mathbf{K},\mathbf{V})}\|\mathcal{F}(\mathbf{X}+\boldsymbol{\Delta}_\mathbf{x}; \hat{\mathbf{W}}) - \mathcal{F}(\mathbf{X}; \mathbf{W})\|^2$ if $\mathbf{Y}$ is unavailable);
7:      $(\mathbf{K}, \mathbf{V}) \leftarrow (\mathbf{K}, \mathbf{V}) - \eta\mathbf{g}$;
8: **end for**
9: **return** $(\mathbf{K}, \mathbf{V})$.

---

where $\ell(\cdot, \cdot)$ is a loss function. In cases where labels of $\mathbf{X}$ are unavailable, we can learn the compensation pool by minimizing the reconstruction loss:

$$\min_{\mathbf{K},\mathbf{V}} \sum_{(\mathbf{X},\cdot)\in\mathcal{D}} \|\mathcal{F}(\mathbf{X} + \boldsymbol{\Delta}_\mathbf{x}; \hat{\mathbf{W}}) - \mathcal{F}(\mathbf{X}; \mathbf{W})\|^2. \quad (7)$$

For NLP tasks, inputs (i.e., sentences) are sequences of discrete tokens, making direct modification of inputs infeasible. To deal with this issue, input compensation is operated in the input embedding space. Let $\mathbf{H}_\mathbf{x} \in \mathbb{R}^{N \times d_e}$ be the embeddings extracted by the input embedding layer of the pruned LLM. Similar to (5), we construct the input compensation as $\boldsymbol{\Delta}_\mathbf{x} = \text{softmax}\left(\frac{\mathbf{H}_\mathbf{x}\mathbf{K}}{\sqrt{d_e}}\right)\mathbf{V}$. The input embeddings are then adjusted as $\mathbf{H} + \boldsymbol{\Delta}_\mathbf{x}$ and we learn the compensation pool by minimizing the reconstruction loss of the last hidden states.

## 5. Experiments

### 5.1. Experiments on Image Classification

**Datasets.** We conduct image classification experiments on ten datasets: CIFAR100 (Krizhevsky & Hinton, 2009), Flowers (Nilsback & Zisserman, 2008), Food (Bossard et al., 2014), EuroSAT (Helber et al., 2019), SUN (Xiao et al., 2016), DTD (Cimpoi et al., 2014), UCF (Soomro et al., 2012), SVHN (Netzer et al., 2011), OxfordPets (Jawahar et al., 2012) (denoted by Pets), and RESISC45 (Cheng et al., 2017) (denoted by RESISC). A summary of the datasets is in Table 10 of Appendix B.6.

**Implementation Details.** We adopt CLIP ViT-B/32 and ViT-B/16 (Radford et al., 2021) as the base models, whose pruned image encoder is used as the encoder of IC. We initialize the $\mathbf{K}$ and $\mathbf{V}$ by the standard normal distribution and train the compensation pool for 30 epochs using the SGD optimizer with a learning rate of 40 and momentum

Table 1: Testing accuracy on image classification tasks using CLIP ViT-B/32.

| | Sparsity | #Nonzero Params | CIFAR100 | Flowers | Food | EuroSAT | SUN | UCF | SVHN | Pets | DTD | RESISC | **Avg** |
|---|---|---|---|---|---|---|---|---|---|---|---|---|---|
| Magnitude | 50% | 110M | 33.9 | 26.1 | 34.2 | 45.6 | 30.8 | 35.4 | 45.3 | 38.7 | 27.9 | 55.4 | 37.3 |
| Magnitude | 48% | 112M | 43.4 | 30.9 | 44.5 | 61.3 | 36.3 | 43.9 | 56.7 | 48.9 | 31.4 | 64.8 | 46.2 |
| Magnitude + IC | 50% | 112M | 73.0 | 62.9 | 72.4 | 96.5 | 48.9 | 63.1 | 94.4 | 69.2 | 44.1 | 87.1 | **71.2** |
| Wanda | 50% | 110M | 75.0 | 56.4 | 74.1 | 95.2 | 50.8 | 59.7 | 91.8 | 57.6 | 43.4 | 84.4 | 68.9 |
| Wanda | 48% | 112M | 77.6 | 63.9 | 77.5 | 95.9 | 55.2 | 65.0 | 89.3 | 65.1 | 45.7 | 87.9 | 72.3 |
| Wanda + IC | 50% | 112M | 80.1 | 76.4 | 80.4 | 97.9 | 54.7 | 69.1 | 96.1 | 77.5 | 49.8 | 91.6 | **77.4** |
| SparseGPT | 50% | 110M | 83.3 | 69.1 | 81.6 | 97.9 | 58.0 | 68.5 | 93.7 | 59.4 | 48.2 | 89.8 | 74.9 |
| SparseGPT | 48% | 112M | 84.5 | 72.4 | 82.7 | 97.9 | 60.4 | 71.7 | 91.2 | 66.3 | 50.2 | 91.3 | 76.9 |
| SparseGPT + IC | 50% | 112M | 82.9 | 76.2 | 83.1 | 98.2 | 57.2 | 71.0 | 96.7 | 79.7 | 53.8 | 92.9 | **79.2** |
| Magnitude | 4:8 | 110M | 49.0 | 25.9 | 36.5 | 45.1 | 32.8 | 37.8 | 60.8 | 45.3 | 27.1 | 60.2 | 42.1 |
| Magnitude + IC | 4:8 | 112M | 72.9 | 62.4 | 72.1 | 96.5 | 48.2 | 62.9 | 94.3 | 68.3 | 44.8 | 87.4 | **71.0** |
| Wanda | 4:8 | 110M | 60.9 | 30.5 | 59.2 | 83.1 | 37.2 | 43.2 | 74.4 | 47.0 | 30.9 | 68.7 | 53.5 |
| Wanda + IC | 4:8 | 112M | 76.9 | 71.4 | 77.3 | 97.2 | 50.2 | 64.2 | 95.2 | 75.3 | 51.1 | 89.5 | **74.8** |
| SparseGPT | 4:8 | 110M | 80.2 | 55.7 | 79.6 | 96.6 | 52.3 | 61.4 | 85.8 | 58.5 | 42.3 | 86.6 | 69.9 |
| SparseGPT + IC | 4:8 | 112M | 81.8 | 72.6 | 81.6 | 98.1 | 51.7 | 65.8 | 96.6 | 78.1 | 49.1 | 92.2 | **76.8** |
| Magnitude | 2:4 | 110M | 30.7 | 11.2 | 19.8 | 42.7 | 19.4 | 23.3 | 27.5 | 25.8 | 15.6 | 35.3 | 25.1 |
| Magnitude + IC | 2:4 | 112M | 78.6 | 66.5 | 77.9 | 97.4 | 53.9 | 67.5 | 96.1 | 70.9 | 44.3 | 90.2 | **74.3** |
| Wanda | 2:4 | 110M | 39.6 | 14.5 | 35.1 | 46.1 | 21.7 | 25.5 | 42.4 | 25.2 | 20.9 | 41.4 | 31.2 |
| Wanda + IC | 2:4 | 112M | 80.6 | 74.5 | 80.7 | 97.7 | 54.9 | 68.1 | 96.5 | 74.1 | 51.4 | 91.6 | **77.0** |
| SparseGPT | 2:4 | 110M | 75.5 | 40.2 | 73.0 | 94.3 | 44.5 | 52.6 | 61.3 | 45.7 | 33.5 | 81.6 | 60.2 |
| SparseGPT + IC | 2:4 | 112M | 82.2 | 79.4 | 82.6 | 98.4 | 57.7 | 69.9 | 96.8 | 76.4 | 54.3 | 92.5 | **79.0** |

Table 2: Testing accuracy on image classification tasks using CLIP ViT-B/16.

| | Sparsity | #Nonzero Params | CIFAR100 | Flowers | Food | EuroSAT | SUN | UCF | SVHN | Pets | DTD | RESISC | **Avg** |
|---|---|---|---|---|---|---|---|---|---|---|---|---|---|
| Magnitude | 50% | 109M | 76.9 | 56.5 | 78.3 | 90.7 | 51.2 | 65.6 | 95.3 | 62.9 | 42.8 | 82.1 | 70.2 |
| Magnitude | 47% | 111M | 80.9 | 64.6 | 82.1 | 94.4 | 56.5 | 70.1 | 95.9 | 68.5 | 47.3 | 86.9 | 74.7 |
| Magnitude + IC | 50% | 111M | 82.9 | 86.7 | 84.7 | 97.6 | 60.1 | 75.1 | 97.1 | 82.5 | 61.1 | 92.8 | **82.1** |
| Wanda | 50% | 109M | 84.1 | 78.1 | 85.5 | 97.6 | 59.5 | 68.9 | 96.9 | 72.7 | 51.8 | 91.2 | 78.6 |
| Wanda | 47% | 111M | 85.6 | 81.7 | 86.8 | 98.0 | 62.8 | 71.7 | 96.8 | 76.8 | 55.3 | 92.9 | 80.8 |
| Wanda + IC | 50% | 111M | 86.2 | 82.8 | 87.8 | 98.4 | 63.8 | 75.5 | 97.6 | 83.6 | 63.5 | 94.7 | **83.4** |
| SparseGPT | 50% | 109M | 87.2 | 80.2 | 88.1 | 98.0 | 63.8 | 73.8 | 97.0 | 75.6 | 56.4 | 93.7 | 81.4 |
| SparseGPT | 47% | 111M | 87.7 | 82.3 | 88.8 | 98.3 | 63.4 | 76.2 | 97.0 | 80.4 | 59.8 | 94.6 | 82.9 |
| SparseGPT + IC | 50% | 111M | 86.1 | 86.0 | 87.9 | 98.4 | 64.4 | 76.2 | 97.6 | 85.2 | 66.4 | 95.0 | **84.3** |
| Magnitude | 4:8 | 109M | 75.8 | 52.0 | 75.4 | 89.6 | 50.0 | 61.4 | 77.4 | 68.8 | 41.4 | 79.1 | 67.1 |
| Magnitude + IC | 4:8 | 111M | 81.5 | 84.2 | 83.2 | 97.5 | 57.6 | 72.5 | 96.9 | 81.6 | 54.4 | 91.9 | **80.1** |
| Wanda | 4:8 | 109M | 78.6 | 63.2 | 81.4 | 96.0 | 50.6 | 61.3 | 78.2 | 69.5 | 42.8 | 87.7 | 70.9 |
| Wanda + IC | 4:8 | 111M | 84.7 | 82.1 | 86.8 | 98.4 | 60.6 | 74.5 | 97.4 | 82.0 | 61.3 | 94.3 | **82.2** |
| SparseGPT | 4:8 | 109M | 85.1 | 74.0 | 87.0 | 95.6 | 60.4 | 69.8 | 72.6 | 78.2 | 50.5 | 93.7 | 76.7 |
| SparseGPT + IC | 4:8 | 111M | 84.7 | 84.9 | 87.1 | 98.3 | 60.9 | 74.9 | 97.5 | 83.6 | 62.2 | 94.4 | **82.9** |
| Magnitude | 2:4 | 109M | 68.3 | 39.9 | 64.5 | 78.6 | 39.4 | 53.2 | 95.3 | 56.9 | 32.9 | 67.6 | 59.7 |
| Magnitude + IC | 2:4 | 111M | 83.9 | 83.8 | 85.9 | 97.9 | 61.4 | 75.7 | 97.4 | 84.0 | 57.3 | 93.3 | **82.1** |
| Wanda | 2:4 | 109M | 71.5 | 46.4 | 71.0 | 89.7 | 40.6 | 50.4 | 96.2 | 61.4 | 33.5 | 75.2 | 63.6 |
| Wanda + IC | 2:4 | 111M | 84.9 | 87.2 | 86.9 | 98.2 | 62.7 | 75.4 | 97.4 | 84.3 | 59.2 | 94.1 | **83.0** |
| SparseGPT | 2:4 | 109M | 82.8 | 66.2 | 82.8 | 94.4 | 53.6 | 63.0 | 97.2 | 74.2 | 43.1 | 90.2 | 74.7 |
| SparseGPT + IC | 2:4 | 111M | 85.6 | 89.6 | 87.7 | 98.4 | 63.9 | 76.9 | 97.6 | 84.9 | 61.2 | 94.9 | **84.1** |

of 0.9. The mini-batch size is 128. Following (Bahng et al., 2022), $\mathbf{v}_i$ is learnable padding pixels on all sides, where the padding size is set to 30. The rank $r$ is chosen as 32 and a sensitivity analysis is provided in Section 6. We evaluate two types of sparsity: unstructured sparsity and structured m:n sparsity (Mishra et al., 2021) (4:8 and 2:4), i.e., at most m out of every n contiguous weights to be non-zero.

**Baselines.** The proposed IC can be integrated into any existing pruning methods. To verify its effectiveness, we consider three pruning methods: (i) Magnitude Pruning (Han et al., 2015) which discards weights based on their magnitudes; (ii) Wanda (Sun et al., 2024) designs

a scoring metric as the weight magnitudes multiplied by the corresponding input activations on a per-output basis; (iii) SparseGPT (Frantar & Alistarh, 2023) which adjusts the unpruned weights by solving a layer-wise reconstruction problem using a second-order optimizer. SparseGPT is a weight compensation method, while Magnitude and Wanda design a scoring metric for pruning without updating weights. For all methods, the base models are fully finetuned on the training set of all tasks before pruning.

**Results.** Tables 1 and 2 show the testing accuracy on ten image classification tasks using CLIP ViT-B/32 and ViT-B/16, respectively. As can be seen, IC consistently

brings large improvements to existing pruning methods in both unstructured (sparsity=50%) and structured (4:8 and 2:4) cases. Specifically, compared with Magnitude, Magnitude + IC achieves improvements of 25% and 10% on ViT-B/32 and ViT-B/16, respectively; Compared with Wanda, Wanda + IC has improvements of about 2.5%; Compared with SparseGPT, SparseGPT + IC performs better by an improvement of 2% on ViT-B/32. The large improvements contributed by IC verify that the learned compensation pool is effective in constructing input compensation for the pruned models. Moreover, SparseGPT + IC consistently performs the best, demonstrating that combining both weight compensation and input compensation is more desirable. We can also observe that unstructured pruning (sparsity=50%) achieves higher accuracy than structured pruning (sparsity=4:8), which is aligned with findings in previous works (Sun et al., 2024; Frantar & Alistarh, 2023; Zhang et al., 2024).

In our IC framework, as the encoder is a submodule of the pruned model, the additional parameters are only from the very small compensation pool (only 2.3M). To verify that IC's improvements over baselines are not due to these additional parameters, we reduce the sparsity of baseline methods to increase the number of nonzero parameters. As shown in Tables 1 and 2, when using the same number of non-zero parameters, combining Magnitude + IC, Wanda + IC, and SparseGPT + IC consistently achieve higher average accuracy than Magnitude, Wanda, and SparseGPT, respectively.

## 5.2. Experiments on Natural Language Processing

**Models and Datasets.** We evaluate IC on the LLaMA model family, i.e., LLaMA-1 (7B) (Touvron et al., 2023a), LLaMA-2 (7B) (Touvron et al., 2023b), and LLaMA-3.1 (8B) (Meta, 2024). Following (Sun et al., 2024; Frantar & Alistarh, 2023), 128 sequences sampled from the first shard of the C4 dataset (Raffel et al., 2020) are used as training data. We evaluate the pruned models on two types of tasks: (i) language modeling task which evaluates the perplexity on the held-out validation data of WikiText-2 (Merity et al., 2016); and (ii) seven zero-shot tasks include BoolQ (Clark et al., 2019), RTE (Wang, 2018), HellaSwag (Zellers et al., 2019), WinoGrande (Sakaguchi et al., 2021), ARC-easy/challenging (Clark et al., 2018), and OpenbookQA (Mihaylov et al., 2018)) from the EleutherAI LM Harness package (Gao et al., 2024). As LLMs contain billions of parameters, to make pruned models more compressed, we follow (Yin et al., 2024) and focus on the unstructured sparsity of 70% case. Implementation details are in Appendix B.4.

**Results on Language Modeling Task.** Table 3 shows the WikiText validation perplexity. As can be seen, IC consis-

Table 3: WikiText validation perplexity of pruned LLaMA family of models.

|  | Sparsity | LLaMA-1 (7B) | LLaMA-2 (7B) | LLaMA-3.1 (8B) |
|---|---|---|---|---|
| Magnitude | 70% | 48432 | 52457 | 3483567 |
| Magnitude + IC | 70% | **19678** | **8585** | **33194** |
| Wanda | 70% | 85.0 | 74.4 | 99.7 |
| Wanda + IC | 70% | **56.5** | **67.0** | **80.1** |
| SparseGPT | 70% | 26.8 | 24.7 | 38.8 |
| SparseGPT + IC | 70% | **17.7** | **18.3** | **27.5** |
| Wanda + OWL | 70% | 24.6 | 30.9 | 70.7 |
| Wanda + OWL + IC | 70% | **19.1** | **24.5** | **63.1** |

tently brings a significant improvement to existing pruning methods, verifying the effectiveness of compensating inputs for pruned LLMs. For example, SparseGPT + IC achieves a perplexity improvement of 6.0 over SparseGPT on all three LLaMA family of models, while Wanda + IC outperforms Wanda by a large margin of 7.0 on all three LLMs. Although Magnitude performs much worse, Magnitude + IC still effectively reduces the perplexity by over 60%.

**Results on Zero-shot Tasks.** Table 4 shows the testing accuracy of seven zero-shot tasks on the LLaMA family of models. As can be seen, IC consistently brings a noticeable improvement (averaged over all tasks) to all existing pruning methods. For example, Wanda + IC outperforms Wanda on LLaMA-3.1-8B, LLaMA-2-7B, and LLaMA-1-7B by margins of $1.09\%$, $1.73\%$, and $0.4\%$, respectively, indicating that the learned compensation pool can be effectively used to construct input compensation for pruned models without any weight update. Moreover, SparseGPT + IC consistently achieves the highest accuracy for all models, showing that learning $\Delta_{\mathbf{x}}$ and $\Delta_{\mathbf{w}}$ are complementary and thus can be combined together for boosting performance.

## 5.3. Experiments on Image Generation

**Experimental Setting.** We evaluate IC on Denoising Diffusion Probability Models (DDPM) (Ho et al., 2020). Following (Fang et al., 2023), the CIFAR-10 dataset (with the image size of $32 \times 32$) (Krizhevsky & Hinton, 2009) and the off-the-shelf DDPM from (Ho et al., 2020) are used. $\mathbf{K}$ is initialized with zero and $\mathbf{V}$ is initialized randomly by a normal distribution with a standard deviation of $0.01$, where the rank $r$ is set to $128$. We train $\mathbf{K}$ and $\mathbf{V}$ using the Adam optimizer (Kingma & Ba, 2015) with a learning rate of $0.002$ over 100K steps. The mini-batch size is set to 128. The identity function is used as the encoder of IC to keep more original image information, which is crucial for image generation. Following (Fang et al., 2023), we focus on the sparsity of 30% case and compare IC with three pruning methods: Magnitude Pruning (Han et al., 2015), Taylor Pruning (Molchanov et al., 2022), and Diff-Pruning (Fang et al., 2023).

**Results.** Table 5 shows the Frechet Inception Distance (FID) (Heusel et al., 2017). As can be seen, IC consis-

Table 4: Testing accuracy of zero-shot tasks using LLaMA family of models.

| | | Sparsity | BoolQ | RTE | HellaSwag | WinoGrande | ARC-e | ARC-c | OBQA | **Avg** |
|---|---|---|---|---|---|---|---|---|---|---|
| **LLaMA-1 (7B)** | Magnitude | 70% | 38.29 | 52.71 | 25.62 | 51.14 | 26.64 | 19.71 | 11.60 | 32.24 |
| | Magnitude + IC | 70% | 55.99 | 52.35 | 25.33 | 48.38 | 25.93 | 21.93 | 15.00 | **34.99** |
| | Wanda | 70% | 57.16 | 54.87 | 28.73 | 50.91 | 32.15 | 18.86 | 13.80 | 36.64 |
| | Wanda + IC | 70% | 59.60 | 53.07 | 28.80 | 52.01 | 34.55 | 18.86 | 12.40 | **37.04** |
| | SparseGPT | 70% | 63.43 | 56.32 | 33.89 | 58.96 | 44.07 | 23.63 | 17.80 | 42.58 |
| | SparseGPT + IC | 70% | 66.06 | 54.87 | 37.47 | 60.06 | 48.40 | 25.51 | 18.20 | **44.37** |
| **LLaMA-2 (7B)** | Magnitude | 70% | 37.95 | 53.07 | 25.95 | 49.25 | 27.74 | 22.78 | 16.80 | 33.36 |
| | Magnitude + IC | 70% | 42.57 | 52.35 | 25.77 | 49.33 | 25.84 | 22.10 | 16.20 | **33.45** |
| | Wanda | 70% | 46.09 | 52.71 | 27.86 | 51.14 | 30.05 | 18.09 | 11.80 | 33.96 |
| | Wanda + IC | 70% | 58.01 | 52.71 | 27.91 | 50.28 | 29.80 | 19.54 | 11.60 | **35.69** |
| | SparseGPT | 70% | 65.75 | 53.07 | 33.47 | 57.06 | 43.73 | 22.35 | 17.40 | 41.83 |
| | SparseGPT + IC | 70% | 65.11 | 52.71 | 36.50 | 57.85 | 49.33 | 24.49 | 17.60 | **43.37** |
| **LLaMA-3.1 (8B)** | Magnitude | 70% | 37.83 | 52.71 | 26.16 | 49.33 | 26.09 | 20.14 | 14.60 | 32.41 |
| | Magnitude + IC | 70% | 37.83 | 53.79 | 25.71 | 49.88 | 25.21 | 22.78 | 15.20 | **32.91** |
| | Wanda | 70% | 56.27 | 52.71 | 27.51 | 47.83 | 32.20 | 17.66 | 13.00 | 35.31 |
| | Wanda + IC | 70% | 61.74 | 52.71 | 27.75 | 49.25 | 33.25 | 17.92 | 12.20 | **36.40** |
| | SparseGPT | 70% | 67.71 | 52.71 | 33.60 | 56.20 | 43.14 | 21.08 | 16.40 | 41.55 |
| | SparseGPT + IC | 70% | 67.71 | 54.15 | 34.25 | 57.62 | 46.63 | 22.78 | 15.60 | **42.68** |

Table 5: FID of pruned DDPMs on CIFAR-10.

| | Sparsity | FID |
|---|---|---|
| Magnitude | 30% | 5.48 |
| Magnitude + IC | 30% | **5.31** |
| Taylor Pruning | 30% | 5.56 |
| Taylor Pruning + IC | 30% | **5.21** |
| Diff-Pruning | 30% | 5.29 |
| Diff-Pruning + IC | 30% | **5.15** |

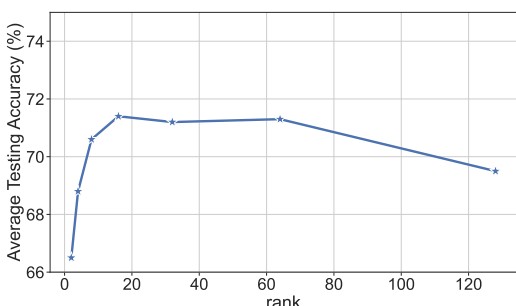

Figure 2: Performance of Magnitude + IC with different ranks on image classification tasks using CLIP ViT-B/32.

tently improves the existing pruning methods, demonstrating the effectiveness of compensating inputs for pruned LLMs. For instance, Taylor Pruning+IC achieves an FID improvement of 0.35 compared to Taylor Pruning. Similarly, Diff-Pruning+IC outperforms Diff-Pruning by 0.14.

# 6. Analysis

In this section, we conduct empirical analyses to investigate the key components of IC, including rank $r$, sparsity, computation cost, sparse retraining, and input-dependent compensation. We adopt the experimental setting used in Section 5.1 with CLIP ViT-B/32.

**Sensitivity of Rank.** We conduct experiments to study the sensitivity of rank $r$ to the performance of Magnitude + IC, where $r$ is chosen from $\{2, 4, 8, 16, 32, 64, 128\}$. Figure 2 shows the testing accuracy with different ranks (detailed results are shown in Table 8 of Appendix B.3). As can be seen, a very small rank (e.g., 2) is not desirable. When the rank is small ($\leq 16$), increasing the rank leads to better performance. A very large rank (e.g., 128) contains more parameters but does not contribute to better performance. In

practice, we can choose the rank $\in [16, 32]$.

**Sensitivity of Sparsity.** We study the performance of Magnitude + IC with different sparsities. Figure 3 shows the trend of testing accuracy (averaged over ten tasks) w.r.t. sparsity (detailed results are shown in Table 6 of Appendix B.1). As can be seen, when the sparsity is high ($\geq 40\%$), Magnitude + IC significantly outperforms Magnitude; When the sparsity is low ($\leq 20\%$), Magnitude + IC and Magnitude perform comparably. In practice, a high sparsity is more desirable for pruning in order to reduce the model size; Thus, IC is practically useful for enhancing pruned models.

**Computation Cost.** Compared with existing pruning methods, the proposed IC needs two extra components (encoder and compensation pool) for adjusting the inputs. As the encoder is either a submodule of the pruned model or an identity function and the compensation pool is very small (only 2.3M in experiments), the extra storage is negligible and the computation cost is small. For example, in the

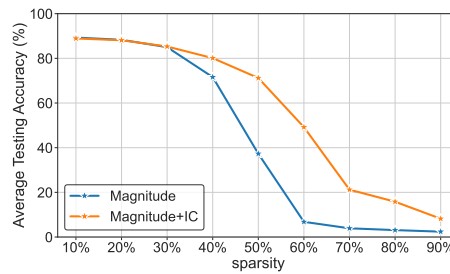

Figure 3: Performance of Magnitude and Magnitude + IC with different sparsities on image classification tasks using CLIP ViT-B/32.

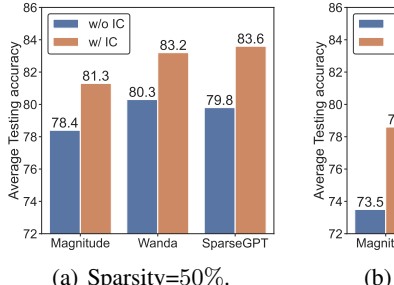
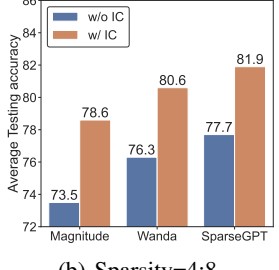

(a) Sparsity=50%.                    (b) Sparsity=4:8.

Figure 4: Testing accuracy (averaged over ten image classification tasks) using CLIP ViT-B/32 with sparse retraining

image classification experiment with CLIP ViT-B/32, compared with Magnitude, computation cost of Magnitude + IC increases by only 1% (from 305G to 309G FLOPs) and inference speed drops slightly from 665 to 535 images/second. Note that testing accuracy increases largely (Table 1), verifying that such additional computation cost is worthwhile.

**Sparse Retraining with IC.** In Section 5.1, we combine IC with three pruning methods without sparse retraining (i.e., retraining the sparse model following the pruning step), which can approach the performance of the dense model. We conduct experiments to investigate whether IC is beneficial to pruning methods with sparse retraining. We retrain retained parameters on the training data for 3 epochs using the AdamW optimizer with a learning rate of 0.000001 and weight decay of 0.01. Figure 4 shows the testing accuracy (averaged over ten tasks) of pruning methods w/ or w/o IC when sparse retraining is applied (detailed results are in Table 7 of Appendix B.2). As can be seen, IC consistently boosts existing pruning methods when sparse retraining is applied. Moreover, sparse retraining achieves higher accuracy than those without retraining (Table 1).

**Input-dependent vs. Input-independent Compensation.** The design of our IC ensures the compensation $\Delta_{\mathbf{x}}$ is input-aware. A straightforward variant of IC is learning a globally shared (i.e., input-independent) compensation $\Delta$ for all inputs. We conduct experiments to investigate the effectiveness of our input-dependent mechanism. Figure 5 shows the testing accuracy (averaged over ten tasks). As can be seen, IC performs much better than the input-independent variant, demonstrating that input-dependent compensation is

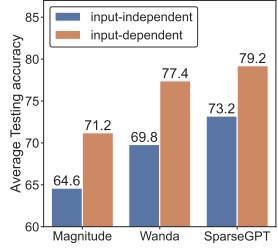
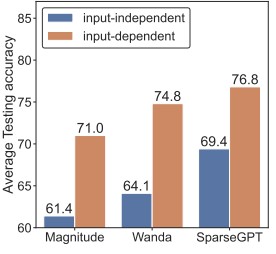

(a) Sparsity=50%.                    (b) Sparsity=4:8.

Figure 5: Testing accuracy (averaged over ten image classification tasks) of IC and an input-independent variant using CLIP ViT-B/32

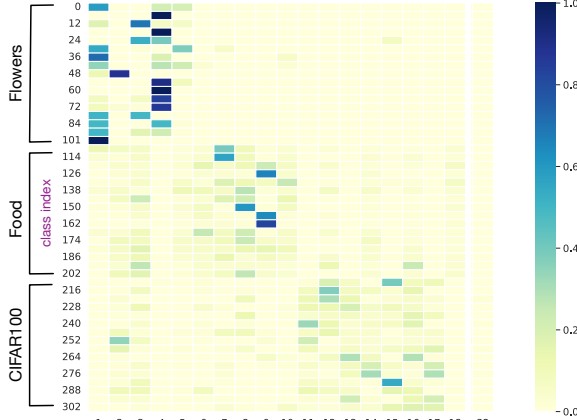

Figure 6: Distribution of attention weights on image classification tasks using CLIP ViT-B/32.

more effective in reducing the error caused by the removed weights.

**Visualization.** In Section 5.1, we learn a compensation pool with $r = 32$ for ten image classification tasks, i.e., $\Delta_{\mathbf{x}}$ is a weighted combination of 32 $\mathbf{v}_i$'s. Here, we study whether different tasks lead to different preferences for $\mathbf{v}_i$'s. Figure 6 shows the average attention weights between $\mathbf{v}_i$ and testing samples belonging to different classes of three tasks (Flowers, Food, CIFAR100) (other tasks are not shown due to limited space). As can be seen, samples from Flowers prefer $\{\mathbf{v}_1, \ldots, \mathbf{v}_5\}$; samples from Food prefer $\{\mathbf{v}_6, \ldots, \mathbf{v}_{10}\}$; samples from CIFAR100 prefer $\{\mathbf{v}_{11}, \ldots, \mathbf{v}_{17}\}$.

## 7. Conclusion

In this paper, we proposed input compensation (IC) for enhancing pruned models by adjusting the inputs to compensate for the error caused by the pruned weights. A pool of multiple compensations is learned to construct input-dependent compensations. IC is designed in the input space while existing pruning methods are designed in the parameter space. Hence, IC can be integrated into any existing pruning methods. Extensive experiments on NLP and CV verify that IC largely enhances pruned models.

## Impact Statement

This paper presents work whose goal is to advance the field of Machine Learning. There are many potential societal consequences of our work, none of which we feel must be specifically highlighted here.

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

# Appendix

## A. Additional Related Works on Foundation Models

Foundation Models are large pre-trained models designed to serve as base models for various downstream tasks. These models are typically trained on a large amount of data and contain massive of parameters. Notable examples include Large Language Models (LLMs) like LLaMA series (Touvron et al., 2023a;b; Meta, 2024), which have promising performance in natural language processing tasks such as text generation (Li et al., 2024; Zhang et al., 2023), understanding (Guo et al., 2024; Fan & Hunter, 2023), and reasoning (Wei et al., 2022; Yu et al., 2024). In the realm of computer vision (CV), models like CLIP (Contrastive Language-Image Pretraining) (Radford et al., 2021) use multimodal learning to bridge textual and visual information, enhancing various CV tasks such as image classification (Radford et al., 2021), image captioning and visual question answering (Li et al., 2022; 2023a). Additionally, diffusion models like DDPM (Ho et al., 2020), Stable Diffusion (Rombach et al., 2022), and SDXL (Podell et al., 2024) have revolutionized image generation by employing a process of gradually transforming noise into images, showing the diverse applications of foundation models in creative applications.

## B. Additional Experimental Results

### B.1. Sensitivity of Sparsity

Table 6 shows the testing accuracy of Magnitude and Magnitude + IC with different sparsities. We can see that Magnitude + IC significantly outperforms Magnitude when the sparsity is high ($\geq 40\%$). In practice, a high sparsity is more desirable for pruning to reduce the model size. Hence, IC is practically useful for boosting the performance of pruned models.

Table 6: Performance of Magnitude and Magnitude + IC with different sparsities.

| Sparsity | IC | CIFAR100 | Flowers | Food | EuroSAT | SUN | UCF | SVHN | Pets | DTD | RESISC | **Avg** |
|---|---|---|---|---|---|---|---|---|---|---|---|---|
| 10% | ✗ | 88.2 | 97.7 | 89.1 | 98.8 | 73.6 | 86.0 | 97.1 | 91.6 | 74.6 | 96.0 | **89.3** |
| 10% | ✓ | 87.7 | 97.6 | 88.8 | 98.5 | 73.3 | 85.1 | 97.0 | 91.7 | 73.8 | 96.0 | 88.9 |
| 20% | ✗ | 87.5 | 96.5 | 88.4 | 98.6 | 72.9 | 84.1 | 96.9 | 90.4 | 72.8 | 95.4 | **88.3** |
| 20% | ✓ | 87.2 | 96.3 | 88.1 | 98.5 | 72.6 | 83.2 | 96.9 | 90.4 | 72.2 | 95.3 | 88.1 |
| 30% | ✗ | 84.8 | 88.8 | 84.2 | 97.6 | 68.6 | 79.2 | 96.5 | 88.3 | 67.7 | 94.1 | 85.0 |
| 30% | ✓ | 84.9 | 91.0 | 85.3 | 98.3 | 68.5 | 78.9 | 96.7 | 88.3 | 67.0 | 94.2 | **85.3** |
| 40% | ✗ | 71.1 | 57.0 | 70.0 | 93.5 | 54.6 | 65.8 | 93.8 | 73.2 | 49.7 | 87.7 | 71.6 |
| 40% | ✓ | 79.8 | 81.9 | 80.7 | 97.4 | 60.4 | 72.9 | 96.2 | 81.4 | 58.9 | 91.6 | **80.1** |
| 50% | ✗ | 33.9 | 26.1 | 34.2 | 45.6 | 30.8 | 35.4 | 45.3 | 38.7 | 27.9 | 55.4 | 37.3 |
| 50% | ✓ | 73.0 | 62.9 | 72.4 | 96.5 | 48.9 | 63.1 | 94.4 | 69.2 | 44.1 | 87.1 | **71.2** |
| 60% | ✗ | 5.6 | 5.2 | 4.3 | 4.3 | 5.5 | 4.7 | 9.1 | 9.0 | 10.0 | 10.2 | 6.8 |
| 60% | ✓ | 57.5 | 25.0 | 52.0 | 89.4 | 26.1 | 37.0 | 85.6 | 29.0 | 16.5 | 74.1 | **49.2** |
| 70% | ✗ | 1.7 | 2.7 | 2.0 | 13.0 | 0.8 | 2.1 | 7.5 | 2.9 | 2.8 | 3.3 | 3.9 |
| 70% | ✓ | 19.3 | 6.1 | 13.8 | 75.4 | 4.0 | 7.0 | 51.6 | 4.1 | 4.0 | 26.6 | **21.2** |
| 80% | ✗ | 1.0 | 1.0 | 0.8 | 13.0 | 0.3 | 1.4 | 6.5 | 2.8 | 1.7 | 2.4 | 3.1 |
| 80% | ✓ | 6.9 | 5.1 | 5.2 | 68.8 | 1.0 | 3.5 | 38.8 | 3.8 | 2.8 | 22.1 | **15.8** |
| 90% | ✗ | 1.1 | 0.5 | 1.0 | 6.9 | 0.2 | 0.7 | 6.4 | 2.6 | 2.1 | 2.1 | 2.4 |
| 90% | ✓ | 3.5 | 2.7 | 2.2 | 49.5 | 0.5 | 2.1 | 6.7 | 3.5 | 3.4 | 7.5 | **8.2** |

## B.2. Sparse Retraining with IC

We conduct experiments to study the performance of IC when sparse retraining is applied. Table 7 shows the testing accuracy on image classification tasks using CLIP ViT-B/32. As shown, IC brings a significant improvement to existing pruning methods when sparse retraining is used.

Table 7: Testing accuracy on image classification tasks using CLIP ViT-B/32 with sparse retraining.

|  | Sparsity | CIFAR100 | Flowers | Food | EuroSAT | SUN | UCF | SVHN | Pets | DTD | RESISC | **Avg** |
|---|---|---|---|---|---|---|---|---|---|---|---|---|
| Magnitude | 50% | 79.3 | 80.8 | 81.2 | 87.8 | 61.6 | 73.8 | 95.5 | 78.2 | 55.1 | 90.6 | 78.4 |
| Magnitude + IC | 50% | 82.4 | 82.6 | 83.0 | 98.2 | 63.1 | 76.3 | 96.8 | 80.0 | 57.7 | 92.5 | **81.3** |
| SparseGPT | 50% | 80.3 | 85.4 | 83.4 | 83.1 | 63.1 | 74.7 | 96.1 | 82.0 | 58.2 | 91.3 | 79.8 |
| SparseGPT + IC | 50% | 84.5 | 88.3 | 86.0 | 98.4 | 65.7 | 77.5 | 96.8 | 83.3 | 61.5 | 93.9 | **83.6** |
| Wanda | 50% | 81.2 | 84.8 | 83.5 | 88.1 | 62.8 | 73.8 | 96.0 | 81.8 | 59.3 | 92.0 | 80.3 |
| Wanda + IC | 50% | 84.6 | 87.4 | 85.3 | 98.4 | 64.7 | 76.6 | 96.8 | 82.6 | 61.5 | 94.0 | **83.2** |
| Magnitude | 4:8 | 75.9 | 70.4 | 78.7 | 77.0 | 57.1 | 68.1 | 94.9 | 76.1 | 47.9 | 88.9 | 73.5 |
| Magnitude + IC | 4:8 | 81.1 | 76.7 | 81.2 | 97.8 | 59.5 | 72.0 | 96.5 | 78.0 | 51.8 | 91.6 | **78.6** |
| SparseGPT | 4:8 | 79.6 | 80.3 | 82.7 | 80.9 | 60.0 | 70.0 | 95.7 | 81.2 | 55.6 | 90.8 | 77.7 |
| SparseGPT + IC | 4:8 | 83.8 | 84.0 | 84.8 | 98.3 | 62.2 | 74.3 | 96.7 | 82.7 | 58.7 | 93.7 | **81.9** |
| Wanda | 4:8 | 78.9 | 76.9 | 81.4 | 80.3 | 58.0 | 68.7 | 95.6 | 78.6 | 54.6 | 90.2 | 76.3 |
| Wanda + IC | 4:8 | 83.6 | 81.4 | 83.3 | 98.0 | 60.5 | 71.9 | 96.7 | 80.0 | 57.3 | 92.8 | **80.6** |

## B.3. Sensitivity of Rank

Table 8 shows the testing accuracy and number of parameters of $(\mathbf{K}, \mathbf{V})$ with different ranks. We can see that a very small rank (e.g., 2) is not desirable. In practice, we can choose the rank from 16 to 32.

Table 8: Testing accuracy of Magnitude + IC (sparsity=50%) with different ranks on image classification tasks using CLIP ViT-B/32.

| Rank | #Params | CIFAR100 | Flowers | Food | EuroSAT | SUN | UCF | SVHN | Pets | DTD | RESISC | **Avg** |
|---|---|---|---|---|---|---|---|---|---|---|---|---|
| 2 | 0.14M | 70.9 | 51.7 | 69.5 | 94.9 | 47.1 | 58.2 | 93.8 | 57.2 | 38.1 | 83.5 | 66.5 |
| 4 | 0.29M | 73.5 | 56.6 | 71.6 | 95.8 | 47.9 | 60.3 | 94.2 | 61.9 | 40.6 | 85.7 | 68.8 |
| 8 | 0.58M | 73.9 | 62.0 | 72.2 | 96.4 | 49.4 | 62.7 | 94.3 | 65.8 | 42.7 | 86.4 | 70.6 |
| 16 | 1.15M | 73.5 | 62.0 | 72.9 | 96.9 | 49.6 | 63.8 | 94.4 | 69.8 | 44.0 | 87.2 | **71.4** |
| 32 | 2.30M | 73.0 | 62.9 | 72.4 | 96.5 | 48.9 | 63.1 | 94.4 | 69.2 | 44.1 | 87.1 | 71.2 |
| 64 | 4.60M | 73.1 | 64.3 | 73.1 | 96.8 | 50.8 | 64.4 | 94.4 | 65.5 | 42.4 | 87.9 | 71.3 |
| 128 | 9.20M | 73.5 | 56.7 | 72.6 | 96.6 | 49.8 | 62.0 | 94.2 | 62.7 | 40.4 | 87.0 | 69.5 |

## B.4. Implementation Details of NLP Experiments

**Implementation Details.** We randomly initialize $\mathbf{K}$ and $\mathbf{V}$ by a normal distribution with zero mean and standard deviation 0.01, where the rank $r$ is set to 32. We train $\mathbf{K}$ and $\mathbf{V}$ using the AdamW optimizer (Loshchilov & Hutter, 2019) with a learning rate of 0.001 and a linear warmup scheduler over 20 epochs. The mini-batch size is set to 1, with a gradient accumulation of 2. The input embedding layer is used as the encoder of IC.

## B.5. Language Modeling Task with Sparsity=50%

For language modeling experiments in Section 5.2, we focus on sparsity of 70% for making models more compressed. Here, we also follow Wanda (Sun et al., 2024) and conduct experiments to evaluate IC under the 50% sparsity. The table below shows the WikiText validation perplexity of pruned LLaMA family of models (results with $^\dagger$ are reported in the original publication). As can be seen, IC consistently brings a significant improvement to existing pruning methods, verifying the effectiveness of compensating inputs for pruned LLMs again.

Table 9: WikiText validation perplexity of pruned LLaMA family of models.

| | Sparsity | LLaMA-1 (7B) | LLaMA-2 (7B) | LLaMA-3.1 (8B) |
|---|---|---|---|---|
| Magnitude[†] | 50% | 17.29 | 14.89 | - |
| Magnitude | 50% | 17.29 | 14.90 | 134.26 |
| Magnitude + IC | 50% | **14.60** | **12.44** | **80.37** |
| Wanda[†] | 50% | 7.26 | 6.41 | - |
| Wanda | 50% | 7.26 | 6.42 | 9.88 |
| Wanda + IC | 50% | **7.17** | **6.33** | **8.80** |
| SparseGPT[†] | 50% | 7.22 | 6.51 | - |
| SparseGPT | 50% | 7.22 | 6.52 | 9.46 |
| SparseGPT + IC | 50% | **7.07** | **6.38** | **8.51** |

## B.6. Statistics of Image Classification Datasets

Table 10 shows the statistics of the image classification datasets.

Table 10: Summary of ten image classification datasets.

| Dataset | Training Size | Testing Size | #Classes |
|---|---|---|---|
| CIFAR100 (Krizhevsky & Hinton, 2009) | 50,000 | 10,000 | 100 |
| Flowers (Nilsback & Zisserman, 2008) | 4,093 | 2,463 | 102 |
| Food (Bossard et al., 2014) | 50,500 | 30,300 | 101 |
| EuroSAT (Helber et al., 2019) | 13,500 | 8,100 | 10 |
| SUN (Xiao et al., 2016) | 15,888 | 19,850 | 397 |
| DTD (Cimpoi et al., 2014) | 2,820 | 1,692 | 47 |
| UCF (Soomro et al., 2012) | 7,639 | 3,783 | 101 |
| SVHN (Netzer et al., 2011) | 73,257 | 26,032 | 10 |
| Pets (Jawahar et al., 2012) | 2,944 | 3,669 | 37 |
| RESISC (Cheng et al., 2017) | 18,900 | 6,300 | 45 |

