# OpenReview forum: "Enhancing Pruned Models by Input Compensation"
_ICML.cc/2025/Conference — Submitted to ICML 2025_

### Official Review · Reviewer_QATu · 2025-02-18

**Overall Recommendation:** 3

**Summary:**

This paper proposes a method called input compensation (IC) for enhancing pruned models by adjusting the input to compensate for the removed weights. IC is designed in the input space and is orthogonal to existing pruning methods designed in the parameter space. Emprically, IC can be combined with existing pruning methods.

**Claims And Evidence:**

yes.

**Essential References Not Discussed:**

The paper discusses that input compensation in the field of pruning is relatively novel. However, should there not also be a discussion on research regarding input compensation outside the realm of pruning, at least from a methodological perspective?

**Experimental Designs Or Analyses:**

In Section 5 of the paper, there is relevant experimental analysis provided.

**Methods And Evaluation Criteria:**

yes.

**Other Comments Or Suggestions:**

The schematic diagram of the methodology (Figure 1) appears to be somewhat oversimplified and would benefit from a more detailed representation.

**Other Strengths And Weaknesses:**

1. The methodology presented in this article demonstrates input compensation applied to a model that has already been pruned. However, is it feasible to concurrently train the required K and V for "input compensation" during the pruning process itself? This aspect warrants a thorough discussion both methodologically and experimentally.

2. Outside the domain of pruning, should there not also be a discourse on the research pertaining to input compensation, at least from a methodological standpoint?

3. While we observe significant performance enhancements due to input compensation, does it also contribute to an improvement in the degree of sparsity? This is an area that merits further investigation.

**Questions For Authors:**

Please see the "Other Strengths And Weaknesses" part.

**Relation To Broader Scientific Literature:**

The primary contribution of this paper lies in the proposal of a novel method that can be integrated with existing pruning techniques. This method demonstrates the capability to enhance the performance of pruning approaches, particularly under conditions of high sparsity.

**Theoretical Claims:**

There are no theoretical claims.

---

> ### Author Rebuttal · Authors · 2025-03-31
>
> Dear Reviewer QATu,
>
> We sincerely thank you for your positive rating, thoughtful review, and valuable suggestions that have helped us improve our paper.
>
> We have carefully addressed your concerns as follows.
> If you have any other concerns or questions, please let us know. We are more than happy to address them and further improve our work.
>
> Best,
>
> Authors
>
> ---
>
> > **Q1.**
> > Discussion on research regarding input compensation outside the realm of pruning
>
> **A1.** Thank you for this insightful suggestion.
> Input compensation (IC) has been studied in other domains, particularly in control systems and signal processing.
>
> - **Control Systems**: As discussed in the last paragraph of Related Work, IC is a well-established technique in control theory [1-2] to adjust control signals for reducing the influence of potential disturbances.
>
> - **Signal Processing**: IC is also relevant to Pre-emphasis in Signal Processing [3], which modifies input signals to counteract the effects of noise and attenuation that can occur in communication channels, especially in analog transmission systems.
>
> We will discuss these connections more explicitly in our revised paper, which will help us better position our contribution within the landscape of input compensation techniques across different fields.
>
> **References**
>
> [1] Automatic control systems. 1995.
>
> [2] Feedback control of dynamic systems. 2002.
>
> [3] Pre-emphasis and speech recognition. 1995.
>
> ---
>
> > **Q2.**
> > However, is it feasible to concurrently train the required K and V for "input compensation" during the pruning process itself? This aspect warrants a thorough discussion both methodologically and experimentally.
>
> **A2.**
>  Thank you for your insightful suggestion.
>
>  We conducted an ablation study (using ten image classification tasks and the CLIP ViT-B/32 model as in the Section 5.1) to explore this variant of IC.
>  As pruning is a non-differentiable process, IC cannot be trained jointly with the pruning process directly.
>  We instead study the performance of jointly training $(K,V)$ with the retained weights after pruning.
> In our study, we compare two approaches:
> - **Pruning $\to$ IC (separate training)**, where we first train the retained weights and then learn the $(K,V)$.
> - **Pruning $+$ IC (joint training)**, where we jointly train the retained weights and $(K,V)$.
>
> Table R1 (https://www.dropbox.com/scl/fi/n3qcoi1ckwz4x2mlw0vbk/results.pdf?rlkey=8mt0tzz1f1ikzaa1hzuzns0lc&st=ldzypniu&dl=0) shows that the **joint training consistently surpasses separate training** across different sparsity patterns.
> This finding underscores the effectiveness of the joint training strategy, which allows for more cohesive optimization, leading to enhanced performance.
>
> ---
>
> > **Q3.**
> > a discourse on the research pertaining to input compensation
>
> **A3.** See our reply to Q1.
>
> ---
>
> > **Q4.**
> > does IC also contribute to an improvement in the degree of sparsity?
>
> **A4.**
> Thank you for your insightful question.
>
> IC not only enhances performance at a given sparsity level but can also **enable existing pruning methods to achieve higher sparsity without significant performance degradation.**
> As shown in Figure 3, Magnitude+IC with 60% sparsity performs better than Magnitude with 50% sparsity, while Magnitude+IC with 50% sparsity performs comparable to Magnitude with 40% sparsity.
> This means IC effectively allows for an additional 10% of weights to be pruned while still maintaining higher/comparable performance.
>
> Figure 3 also indicates that **IC improves the tradeoff curve between sparsity and performance**.
> For any target performance level, a model with IC can achieve that performance at a higher sparsity level than without IC.
> By enabling higher effective sparsity, IC allows practitioners to deploy smaller models without sacrificing as much performance as traditional pruning methods would require.
>
> We will add the above discussion to the revised paper.
>
> ---
>
> > **Q5.**
> > improve the diagram of the methodology (Figure 1)
>
> **A5.** Thank you for this valuable suggestion.
> We have created a more comprehensive diagram (https://www.dropbox.com/scl/fi/n3qcoi1ckwz4x2mlw0vbk/results.pdf?rlkey=8mt0tzz1f1ikzaa1hzuzns0lc&st=ldzypniu&dl=0) with the following enhancements:
>
> 1. **Detailed Attention Mechanism**: The revised figure explicitly shows how the query-dependent compensation is generated from the query and how it interacts with the key-value pairs in the compensation pool through the attention mechanism.
>
> 2. **Encoder is part of the pruned model**: The revised figure explicitly shows that the encoder is part of the pruned model.
>
> This enhanced diagram will be included in the revised paper to provide readers with a more intuitive understanding of our IC methodology.

---

> > ### Comment · Reviewer_QATu · 2025-04-06
> >
> > Thank you for the authors’ response.I will maintain my current score.

---

> > > ### Author Response · Authors · 2025-04-07
> > >
> > > Thank you for your follow-up comment and for keeping your score positive.
> > >
> > > We are glad that our response has resolved your initial concerns.

---

### Official Review · Reviewer_aNPP · 2025-03-08

**Overall Recommendation:** 3

**Summary:**

The paper proposes an input compensation approach for pruning, which reformulates weight tuning as adaptive input modifications. Specifically, the method begins with the dual problem of weight compensation and approximates input compensation using a pre-trained encoder and attention-based computations. Experimental results demonstrate improvements over existing pruning methods across multiple tasks.

**Claims And Evidence:**

1. The claim regarding efficiency should be reconsidered. In the first paragraph of the Introduction, the authors briefly discuss existing efficiency-focused methods, including distillation, quantization, and pruning. However, pruning itself often requires parameter tuning and can be hardware-unfriendly, particularly in the case of unstructured pruning [1]. Therefore, this argument should be presented with greater caution.

2. The assertion that the proposed method is orthogonal to existing approaches may not be entirely accurate, especially when considering the approximation target. Check more details in Methods and Evaluation Criteria part.

[1] Fang, Gongfan, et al. "Depgraph: Towards any structural pruning." Proceedings of the IEEE/CVF conference on computer vision and pattern recognition. 2023.

**Essential References Not Discussed:**

None

**Experimental Designs Or Analyses:**

## Strength ##

1. The results on language tasks are strong. Since the baselines are specifically designed for LLMs, the comparison appears fair and demonstrates that the proposed method can sometimes outperform weight compensation empirically.

## Weakness ##

1. Since input compensation is a variant of weight compensation, the authors should provide an error analysis for pruning to justify its effectiveness, rather than solely reporting accuracy improvements.

2. The experiments focus on computer vision tasks (image classification and generation), yet the baselines used for comparison—SparseGPT and Wanda—are designed for language models. Given that weight and activation distributions differ across domains, the optimal hyperparameters for these baselines may also vary. This discrepancy raises concerns about the fairness of the comparisons, as the baselines may not be evaluated under their best settings.

3. In the image classification tasks, the CLIP image encoder is used as the encoder in Figure 1. Although it is pruned, the associated computational budget should not be overlooked when compared to standard pruning methods. The authors should provide more details on this aspect to clarify its impact.

**Methods And Evaluation Criteria:**

The rationale behind the effectiveness of input compensation remains unclear. From the perspective of pruning error, input compensation is the dual of weight compensation, meaning both address the same error. In the linear case, they are theoretically equivalent. However, there is no evidence demonstrating why input compensation performs better or how it aids weight compensation—whether due to optimization challenges or other factors. Furthermore, when extending input compensation to nonlinear models, approximation errors arise. Given this, input compensation does not appear to be truly "orthogonal" to existing methods but rather a variant that modifies the optimization parameters.

**Other Comments Or Suggestions:**

None

**Other Strengths And Weaknesses:**

## Strength ##

The paper is well-written, with a clear presentation of the motivation, methodology, and experiments, making it easy to follow.

**Questions For Authors:**

None

**Relation To Broader Scientific Literature:**

None

**Theoretical Claims:**

No theoretical claim.

---

> ### Author Rebuttal · Authors · 2025-04-01
>
> Dear Reviewer aNPP,
>
> Thank you for your time and effort in reviewing our paper.
>
> We have carefully addressed your concerns and hope you are satisfied with our responses.
>
> **If you have any further questions/concerns, please let us know** and we are more than happy to address them.
>
> Best,
>
> Authors
>
> ---
>
> > **Q1.**
> > The claim regarding efficiency should be reconsidered.
>
> **A1.** Thank you for this important point.
> We agree that pruning itself has limitations:
> Though pruning (especially unstructured pruning) reduces parameter count, it may not always improve hardware efficiency since some general-purpose devices cannot efficiently process sparse computations.
>
> We will do the following to improve the writing:
> - Remove the sentence "Quantization requires specialized hardware support, while distillation requires extensive retraining".
> - Provide a more balanced discussion of different efficiency-focused methods.
> - Add a limitation section to discuss the hardware compatibility issues.
>
> We believe these revisions will present a more accurate view of efficiency-focused methods.
>
> ---
>
> > **Q2.**
> > The assertion that the proposed method is orthogonal to existing approaches may not be entirely accurate
> >
> >The rationale behind the effectiveness of input compensation remains unclear.
> >
> >input compensation does not appear to be truly "orthogonal" to existing methods but rather a variant that modifies the optimization parameters.
> >
> > Since input compensation is a variant of weight compensation, the authors should provide an error analysis for pruning to justify its effectiveness, rather than solely reporting accuracy improvements.
>
> **A2.**
> We believe there is a **misunderstanding regarding orthogonality**.
> When we say IC is "orthogonal" to existing pruning methods, we mean:
> - **Different Spaces**: IC operates in the input space by adjusting inputs, while existing methods based on weight compensation operate in the parameter space by adjusting weights.
> - **Complementary Approaches**: Because they work in different spaces, IC can be combined with existing pruning methods (which are based on weight compensation) to improve their performance. Hence, **IC is not a variant of weight compensation.**
>
> **Testing Accuracy is a Reasonable Metric.**
> Note that both IC and weight compensation fundamentally aim to minimize the output deviation caused by weight removal. Testing accuracy directly quantifies this deviation and is the standard evaluation metric in the pruning literature. Hence, we believe testing accuracy is a reasonable metric to reflect the pruning performance.
>
> **Effectiveness of IC.**
> Since IC and weight compensation are complementary approaches, the effectiveness of IC can be verified by comparing "weight compensation + IC" against "weight compensation alone." Our extensive experiments (Tables 1-5) consistently show that SparseGPT+IC outperforms SparseGPT (a weight compensation method) across various sparsity patterns and tasks, confirming the effectiveness of IC.
>
> ---
>
> > **Q3.**
> > fairness of the comparisons
>
> **A3.** Thank you for this concern.
>
> We would like to claim that the comparison is fair due to the following reasons:
>
> - **Careful Hyperparameter Tuning**: (i) Magnitude and Wanda have no hyperparameters. (ii) We have carefully tuned the hyperparameters of SparseGPT specifically for the vision models. Indeed, the performance of SparseGPT is insensitive to hyperparameters (Hessian dampening, and mask selection blocksize).
>
> - **Wanda and SparseGPT are Competitive**: Note that our CV experiments are based on ViT models, which are transformer like LLMs. Hence, Wanda and SparseGPT, which are initially designed for LLMs, are still SOTA baselines for CV tasks. As shown in Tables 1 and 2, Wanda and SparseGPT achieve much higher accuracy than Magnitude Pruning, thus, are very competitive.
>
> - **Comprehensive Evaluation Across Domains**: Besides CV tasks, we also extensively evaluated our method on NLP tasks (Tables 3 and 4), which shows that IC consistently improves performance across both CV and NLP domains.
>
> In the revised paper, we will include the above discussion to address this concern.
>
> ---
>
> > **Q4.**
> > computation cost of the encoder.
>
> **A4.**
> Thank you for this question.
>
> We have discussed the computation cost in our submission (last paragraph, Page 7).
> Our method **increases FLOPs by only 1%** (from 305G to 309G) compared to existing pruning methods. For a detailed breakdown, we provide Table R1 (https://www.dropbox.com/scl/fi/le76r4j7kdq9ixvzzd0rk/results_flops.pdf?rlkey=75b98wu5xaoe3hkqy9mnik6ut&st=cy9k8676&dl=0) to compare Magnitude and Magnitude+IC FLOPs by module.
> **Given the substantial performance improvements (up to 33.9% accuracy increase in Table 1), this minor computational overhead from constructing compensation is worthwhile**.
>
> We will include this detailed analysis in the revised paper.

---

> > ### Comment · Reviewer_aNPP · 2025-04-04
> >
> > Thank you for the authors’ response. I still have two concerns:
> >
> > 1. Regarding orthogonality, while I agree that the proposed method can be combined with weight compensation, if both techniques address the same source of error, it is important to include experiments demonstrating how input compensation and weight compensation interact to reduce errors. Specifically, I would like to see some analysis of pruning error—even on simpler models—to better understand the contribution of each component.
> >
> > 2. The table provided indicates that language processing is the primary computational bottleneck compared to the vision model. However, for image classification tasks, prompts are typically short, and in many cases, they can be cached, even if prompt processing is expensive. Could the authors provide more details about how the computational evaluation was conducted?
> >
> > As these concerns remain unresolved, I will maintain my current rating.
> >
> > ---
> >
> > Update:
> >
> > Thanks for the reply. I will raise my rating to 3.

---

> > > ### Author Response · Authors · 2025-04-07
> > >
> > > Thank you for your further comments.
> > > We address your concerns as follows.
> > >
> > > ---
> > >
> > > > **Q5.**
> > > > ... include experiments demonstrating how input compensation and weight compensation interact to reduce errors. ... some analysis of pruning error.
> > >
> > > **A5.** We understand "pruning error" to refer to the discrepancy between the output of the dense model $\mathcal{F}(\cdot; W)$ and the pruned model $\mathcal{F}(\cdot; \hat{W})$, rather than the parameter distance $\\|W-\hat{W}\\|^2$.
> > > This is because minimizing the output error is the ultimate goal of pruning methods, while minimizing $\\| W-\hat{W}\\|^2$ can be achieved through simple Magnitude Pruning.
> > > (Remarks: If we have misunderstood the definition of the pruning error, please let us know.)
> > >
> > > To address this concern, we analyzed the **KL divergence** $\text{KL}(\mathcal{F}(\cdot; W) \\| \mathcal{F}(\cdot; \hat{W}) )$ between the output probability distributions of the dense model and the pruned model for the image classification task using ViT-B/32.
> > > As shown in Table R4 (https://www.dropbox.com/scl/fi/le76r4j7kdq9ixvzzd0rk/results_flops.pdf?rlkey=75b98wu5xaoe3hkqy9mnik6ut&st=cy9k8676&dl=0), our analysis reveals two insights:
> > >
> > > - **IC provides consistent benefits across methods**: IC consistently reduces KL divergence for existing methods across different sparsity patterns, demonstrating the effectiveness of IC.
> > >
> > > - **Weight and input compensation are complementary**: The combination of weight compensation (SparseGPT) and IC achieves the lowest KL divergence, showing that weight compensation and IC are complementary.
> > >
> > > ---
> > >
> > > > **Q6.**
> > > > for image classification tasks, prompts are typically short, and in many cases, they can be cached, even if prompt processing is expensive. Could the authors provide more details about how the computational evaluation was conducted?
> > >
> > > **A6.** Thank you for this insightful question. We address this concern from two perspectives.
> > >
> > > ### **(1) Prompt caching considerations in CLIP models**
> > >
> > > We clarify how a CLIP model predicts image classes.
> > > 1. Given an image $x$ (for IC, $x\gets x + \Delta_x$), an image encoder $\mathcal{T}\_{\text{image}}$ computes its image embedding $\mathbf{e}\_x = \mathcal{T}\_{\text{image}}(x)$.
> > > 2. For $N$ classes with prompts $\\{\mathbf{p}\_i\\}\_{i=1}^N$, where $\mathbf{p}\_i=\text{``This is a photo of a \\{the i-th class-name\\}''}$ is the prompt for the $i$-th class.
> > > A text encoder $\mathcal{T}\_{\text{text}}$ computes text embeddings $\mathbf{t}\_i=\mathcal{T}\_{\text{text}}(\mathbf{p}\_i), i=1,\dots, N$.
> > > 3. The model predicts the class with highest cosine similarity $\frac{\mathbf{e}_x^\top \mathbf{t}_i}{\\|\mathbf{e}_x\\|\\|\mathbf{t}_i\\|}, i=1,\dots, N$.
> > >
> > > All computational costs are measured by the `FlopAnalyzer` in MMEngine (https://mmengine.readthedocs.io/en/latest/api/generated/mmengine.analysis.FlopAnalyzer.html).
> > >
> > > **We agree that for query-independent prompts, text embeddings at step 2 can be cached.** **However, caching is infeasible for the query-dependent prompts**,
> > > which contain query-specific information (e.g.,
> > > $\mathbf{p}_i^{(x)}=[v_1(x), \dots, v_k(x), \text{This is a photo of a \\{the i-th class name\\}}]$, where $v_1(x), \dots, v_k(x)$ are discrete/continuous tokens depend on $x$). These dynamic prompts have demonstrated superior performance over query-independent prompts in recent work [1-3].
> > >
> > > ### **(2) Lightweight encoder implementation for IC**
> > >
> > > To further address this concern, we demonstrated that IC can be implemented with minimal computational overhead by reusing a very lightweight submodule of the pruned model:
> > >
> > > - **For image classification tasks**:
> > > We conducted an additional experiment by using only the **first convolutional layer of the image encoder of CLIP-ViT-B/32** as the encoder $\mathcal{E}$ of IC (denoted by "IC (conv1)").
> > > As shown in Tables R1 and R2 (attached in the above anonymous link), compared with Magnitude,
> > > Magnitude+IC (conv1)
> > > incurs just **0.1G FLOPs (only 0.03% increase)** while improving accuracy by 68.6% (from 37.3% to 62.9%).
> > >
> > > - **For the language modeling tasks** in Section 5.2: We have adopted only **the input embedding layer of the language model** as the encoder (Line 761).
> > > Table R3 (attached in the above anonymous link) shows that IC adds just **0.75% computational overhead** while reducing the perplexity significantly by more than 5.5 points (Table 3).
> > >
> > > These results demonstrate that IC can achieve large performance improvements with almost no additional computational cost, making it highly practical for real-world applications.
> > >
> > > If you have any further questions/concerns, please **update the previous comment** to let us know and we are more than happy to address them.
> > >
> > > ---
> > >
> > > **References**
> > >
> > > [1] Conditional Prompt Learning for Vision-Language Models. CVPR 2022
> > >
> > > [2] Learning to Prompt for Vision-Language Models. IJCV 2022
> > >
> > > [3] Enhancing CLIP with GPT-4: Harnessing Visual Descriptions as Prompts. ICCV 2023 Workshop
> > >
> > > ---
> > > Update:
> > >
> > > Thank you for raising the score!

---

### Official Review · Reviewer_MNqZ · 2025-03-13

**Overall Recommendation:** 4

**Summary:**

The work, Enhancing Pruned Models by Input Compensation, proposes a new fine-tuning method where, instead of compensating the retained parameters in compressed neural network models, the work introduces input compensation to adjust inputs to compensate the removed parameters and fine-tune the pruned models for classification performance improvement. In particular, the authors introduce a framework with a pre-trained encoder and a learnable compensation pool to learn input compensation. Once the compensation pool is trained, it can be used to fine-tune the pruned models to improve classification performance.

**Claims And Evidence:**

The work provides convincing and supportive framework figures, mathematical equations, and a model-training algorithm to support its claims. This work aims to determine an input compensation such that the output of pruned models is close to the output of the corresponding dense model.

**Essential References Not Discussed:**

The work mentions essential model compression techniques, including pruning, quantization, and knowledge distillation. Then, the work also mentions the reference of prompting for transformer-based models. Finally, the work describes the background of input compensation in control systems. Inspired by it, the authors leverage this idea to model compression.

**Experimental Designs Or Analyses:**

The authors evaluate the proposed framework and other existing frameworks on multiple standard image classification benchmarks such as CIFAR-10, CIFAR-100, and SUN. Therefore, the experimental designs are convincing. Additionally, the experimentation includes varying ablation studies to empirically prove the success of the proposed input compensation framework.

**Methods And Evaluation Criteria:**

The proposed method is the research problem in model compression. Instead of fine-tuning the retained parameters in the pruned model, this work aims to learn input compensation to reduce the performance gap between the pruned and dense model by the proposed framework. The evaluation criteria are clear enough. The authors evaluate the proposed input compensation framework on different datasets and network architectures, including foundation models. Therefore, the evaluation datasets are good enough for this work.

**Other Comments Or Suggestions:**

No extra comments and suggestions for this work.

**Other Strengths And Weaknesses:**

Strengths:

1. The paper is well-written. Readers can easily understand the proposed framework and its differences from the others.
2. The proposed framework could be used for other network architectures, such as transformer-based models.

Weaknesses:

1. It would be nice if the authors could have theoretical claims for the proposed framework.
2. It would be nice if the authors could test their method on large-scale imge classification datasets.

**Questions For Authors:**

I do not have other questions for the authors.

**Relation To Broader Scientific Literature:**

This work's key contributions are highly relevant to prior scientific findings in efficient deep neural networks and neural network architecture optimization for edge device applications. The proposed framework addresses an essential research problem - fine-tuning the pruned model effectively and efficiently - in model compression-related research.

**Theoretical Claims:**

The work does not provide theoretical claims as evidence for the proposed method.

---

> ### Author Rebuttal · Authors · 2025-03-31
>
> Dear Reviewer MNqZ,
>
> We sincerely thank you for your positive rating, thoughtful review, and valuable suggestions that have helped us improve our paper.
> We have carefully addressed your concerns as follows.
>
> If you have any other concerns or questions, please let us know. We are more than happy to address them and further improve our work.
>
> Best,
>
> Authors
>
> ---
>
> > **Q1.**
> > It would be nice if the authors could have theoretical claims for the proposed framework.
>
> **A1.**
> Thank you for your valuable suggestion.
> We agree that strengthening the theoretical foundations would enhance our paper.
> We have provided a theoretical analysis for linear models in Section 4.1, where we show **the duality between input compensation and weight compensation**:
>
> For a linear layer with output $Y = XW$, we demonstrate that if the weight matrix $W$ can be approximated as $W \approx S + AB^\top$, where $S$ is a sparse matrix and $AB^\top$ is a low-rank matrix. This leads to
> $$Y = XW \approx X(S + AB^\top) = XS + (X + XA\hat{B}^\top)S,$$
> where $\hat{B}\equiv B^\top(S^\top S)^{-1}S^\top$.
> This equivalence shows that adjusting the input (adding $XA\hat{B}^\top$ to $X$) can have similar effects as adjusting the weights (adding $AB^\top$ to $S$) in linear models.
>
> In the revised paper, we will strengthen our theoretical analysis by providing a formal theorem and proof for the equivalence between input compensation and weight compensation in linear models.
> However, **extending this analysis to non-linear models is non-trivial** and requires additional theoretical work. We leave this as a future research direction.
>
> ---
>
> > **Q2.**
> > It would be nice if the authors could test their method on large-scale image classification datasets.
>
> **A2.**
> We are grateful for your valuable suggestion.
> We conducted additional experiments on **ImageNet**, a **large-scale** image classification dataset with 1,000 classes and over 1.2 million training images.
> Table R1 (https://www.dropbox.com/scl/fi/8srj7o2yacxmxtg9j75j7/results_on_imagenet.pdf?rlkey=krue8mns0ues2ns9uulcxlnae&st=fhqqqoj5&dl=0) compares the testing accuracy of different methods using CLIP ViT-B/32 .
> As can be seen, **our IC method consistently increases the testing accuracy of existing pruning methods** across different sparsity patterns on ImageNet:
>
> - At unstructured sparsity (50%), IC improves Magnitude Pruning from 19.6% to 41.0% accuracy (+21.4%), Wanda from 38.3% to 51.0% (+12.7%), and SparseGPT from 47.7% to 52.9% (+5.2%).
>
> - For the more challenging structured sparsity patterns (4:8 and 2:4), IC shows even more significant improvements. For example, with the 2:4 pattern, IC improves Magnitude Pruning from 7.2% to 32.5% (+25.3%), Wanda from 11.6% to 36.1% (+24.5%), and SparseGPT from 36.0% to 48.7% (+12.7%).
>
> These results on ImageNet further validate that **our IC method effectively enhances model performance on large-scale image classification tasks**.

---

### Official Review · Reviewer_Wf6n · 2025-03-15

**Overall Recommendation:** 3

**Summary:**

The paper introduces a novel post-pruning algorithm that enhances pruned models by leveraging input compensation (IC) instead of traditional weight updates. This approach is compatible with any pruning method. Through extensive experiments on ViT, LLaMa, and DDPM, the study demonstrates that the proposed attention-based IC design effectively learns and applies input compensation, leading to significant performance improvements across diverse tasks and pruning strategies.

**Claims And Evidence:**

yes

**Essential References Not Discussed:**

No.

**Experimental Designs Or Analyses:**

Input compensation is an interesting and novel approach. However, I believe it requires a more thorough analysis compared to traditional weight update algorithms.

Q1: When comparing pruning methods with and without IC, the baselines include fine-tuning to recover performance loss. Could you provide details on the fine-tuning process for these methods, including the number of epochs and datasets used?
Q2: In the ViT experiments, was ViT fine-tuned separately for each subtask? Additionally, was a single IC trained across all subtasks, or was a unique IC trained for each subtask?
Q3: This method is inspired by the theoretical equivalence between input compensation and weight compensation and is trained extensively to learn IC. In essence, it seems to distill weight compensation into input compensation. However, it remains unclear whether this approach provides an advantage over pruning combined with LoRA fine-tuning. To clarify this, it would be beneficial to conduct experiments using LoRA with a comparable number of additional parameters and directly compare its effectiveness with IC.

**Methods And Evaluation Criteria:**

yes

**Other Comments Or Suggestions:**

None.

**Other Strengths And Weaknesses:**

Strengths:
The paper is well-written and well-organized, with solid experiments demonstrating the effectiveness of IC in post-pruning.

Weaknesses:
As noted in the questions above, the advantage of IC over pruning combined with LoRA remains unclear. A direct comparison would strengthen the paper.
I will raise my score if the authors show their advantage over LoRA + FT.

**Questions For Authors:**

See above questions.

**Relation To Broader Scientific Literature:**

This paper aligns with the research about how data perturbation affects model performance. Perspectively, it relates to prompt tuning and adversarial attacks, as both modify inputs to influence model behavior. Prompt tuning optimizes inputs to guide responses, while adversarial attacks introduce perturbations to manipulate predictions. Unlike adversarial attacks, IC enhances pruned models by compensating for lost weights. Additionally, IC offers an alternative to weight updates in model compression, complementing existing pruning strategies and aligning with efficient adaptation methods like LoRA.

**Theoretical Claims:**

This method is inspired by input compensation in linear models and introduces an equivalence between IC and weight updates in theory. This part appears reasonable. However, real-world models are not purely linear, and there is no theoretical guarantee for the effectiveness of IC in general cases. Since pruning algorithms are typically heuristic, the absence of a formal theoretical guarantee is acceptable in this context.

---

> ### Author Rebuttal · Authors · 2025-03-31
>
> Dear Reviewer Wf6n,
>
> We sincerely thank you for your thoughtful review and valuable suggestions that enhanced our paper.
>
> We have carefully addressed your concerns as follows.
> **If you have any other concerns or questions, please let us know.** We are more than happy to address them and further improve our work.
>
> Best,
>
> Authors
>
> ---
>
> > **Q1.** No theoretical guarantee for the effectiveness of IC in general cases.
>
> **A1.** Thank you for your valuable question.
> We agree that our IC method has a **limitation** that lacks a theoretical guarantee for non-linear models.
> Though we have provided a theoretical analysis for linear models in our paper, **extending this analysis to non-linear models remains a challenging open problem that we leave as a future research direction**.
>
> Despite this theoretical limitation, our approach follows the practical tradition of many successful pruning methods that are primarily heuristic in nature.
> The empirical results across various tasks (image classification, language modeling, and image generation) demonstrate that IC effectively boosts the performance of pruned models in practice. **These consistent performance improvements across different model architectures and tasks provide strong evidence for the practical utility of our method, even in the absence of complete theoretical guarantees for non-linear cases**.
>
> ---
>
> > **Q2.**
> >  Could you provide details on the fine-tuning process for these methods, including the number of epochs and datasets used?
>
> **A2.** Thank you for your question about the fine-tuning process (i.e., sparse retraining).
> For all methods, we followed a **consistent** fine-tuning protocol to ensure fair comparison:
>
> - Datasets: The fine-tuning is performed on the **same** training datasets (i.e., the ten datasets in Section 5.1) used for learning the IC.
>
> - Number of epochs: We retrain the retained parameters for 3 epochs for **all** methods. We observed that performance typically **saturates** after 2 epochs, with minimal gains from additional training.
>
> - Optimizer: For all methods, we adopt the AdamW optimizer with a learning rate of 0.000001 and weight decay of 0.01, and a batch size of 128.
>
> This consistent protocol ensures that any performance improvements observed when combining pruning methods with IC can be attributed to the effectiveness of our approach rather than differences in the fine-tuning process.
>
> ---
>
> > **Q3.** In the ViT experiments, (Q3-A) was ViT fine-tuned separately for each subtask? (Q3-B) Additionally, was a single IC trained across all subtasks, or was a unique IC trained for each subtask?
>
> **A3.** Thank you for your questions.
>
> _(Q3-A)_ **No**, the ViT model was not fine-tuned separately for each subtask. Instead, we fine-tuned a single ViT model across all ten subtasks. This multi-task approach is **more parameter-efficient** as we maintain only one model instead of ten separate models and better reflects real-world deployment scenarios where a single pruned model needs to handle various tasks.
>
> _(Q3-B)_ **Yes**, a single IC was trained across all subtasks. All subtasks share the same compensation pool, which provides two benefits:
> - **Parameter efficiency**: Using a shared compensation pool requires much fewer parameters compared with training separate ICs for each subtask.
> - **Knowledge sharing**: As shown in Figure 6, different subtasks can share the same components of the compensation pool, demonstrating effective knowledge sharing.
>
> ---
>
> > **Q4.**
> > However, it remains unclear whether this approach provides an advantage over pruning combined with LoRA fine-tuning. To clarify this, it would be beneficial to conduct experiments using LoRA with a comparable number of additional parameters and directly compare its effectiveness with IC.
>
>
> **A4.**
> Thank you for your insightful suggestion.
>
> To address your concern, we conducted additional language modeling experiments to compare IC with LoRA fine-tuning on pruned LLMs using approximately the same number of additional parameters as our IC method.
>
> For LoRA implementation, we use a rank of 16 for both LLaMA-1 and LLaMA-2. To maintain **the same number of parameters (only 262K)** as our IC method, LoRA is applied only to the first layer of the LLM, which our ablation studies showed to be the most effective configuration.
>
> Table R1 (https://www.dropbox.com/scl/fi/x9ni5gmmetxkog2dslsac/results_lora.pdf?rlkey=jdzet17nezcsoxtrfxbyr888g&st=1pyx8l6t&dl=0) shows that IC consistently outperforms LoRA when combined with various pruning methods (Magnitude Pruning, Wanda, and SparseGPT) across both LLaMA-1 and LLaMA-2 models with the same parameter budget.
>
> This suggests that, **when using extremely few trainable parameters, IC is more effective than LoRA**.
> We will add this experiment to the revised paper.
>
> ---
>
> > **Q5.**
> > the advantage of IC over pruning combined with LoRA remains unclear.
>
> **A5.**
> See our reply to Q4.

---

> > ### Comment · Reviewer_Wf6n · 2025-04-06
> >
> > Thank you for the clarifications; they have effectively addressed my concerns. Notably, the fact that IC outperforms LoRA highlights the potential advantages of input compensation over weight compensation. I have raised my score.

---

> > > ### Author Response · Authors · 2025-04-07
> > >
> > > Thank you for your further comments and for raising your score.
> > >
> > > We are glad that our reply and additional experiments have resolved your concerns.

---

### Decision · Program_Chairs · 2025-05-01

**Decision:**

Reject

**Comment:**

In the usual NN compression pipeline, the model W is tuned to obtain some W', and then a fine-tuning step is performed on W' to obtain W''. In this paper, they propose to modify _inputs_ with the goal of ensuring that W'' x = W' x', for a modified x'. The paper shows that a small transformer network can learn the transformation x -> x' and achieve an improvement in several tasks. The reviewers all note that the paper is purely empirical, and some justification is provided only for the linear case (where it is very simple).